# AI-first structural identification of pathogenic protein target interfaces

Mihkel Saluri[1¤a], Michael Landreh[1¤b], Patrick Bryant [2]*

1 Department of Microbiology, Tumor and Cell Biology, Karolinska Institutet, Solna, Sweden, 2 Science for Life Laboratory, The Department of Molecular Biosciences, The Wenner-Gren Institute, Stockholm University, Solna, Sweden

¤a Current address: School of Natural Sciences and Health, Tallinn University, Tallinn, Estonia
¤b Current address: Department of Cell and Molecular Biology, Uppsala University, Uppsala, Sweden
* patrick.bryant@scilifelab.se

## Abstract

The risk of pandemics is increasing as global population growth and interconnectedness accelerate. Understanding the structural basis of protein-protein interactions between pathogens and hosts is critical for elucidating pathogenic mechanisms and guiding treatment or vaccine development. Despite 21,064 experimentally supported human-pathogen interactions in the HPIDB, only 52 have resolved structures in the PDB, representing just 0.2%. Advances in protein complex structure prediction, such as AlphaFold, now enable highly accurate modelling of heterodimeric complexes, though their application to host-pathogen interactions, which have distinct evolutionary dynamics, remains underexplored. Here, we investigate the structural protein-protein interaction network between humans and ten pathogens, predicting structures for 9,452 interactions, only 10 of which have known structures. We identify 30 interactions with an expected TM-score ≥0.9, tripling the structural coverage in these networks. A detailed analysis of the Francisella tularensis dihydroprolyl dehydrogenase (IPD) complex with human immunoglobulin kappa constant (IGKC) using homology modelling and native mass spectrometry confirms a predicted 1:2:1 heterotetramer, suggesting potential roles in immune evasion. These findings highlight the transformative potential of structure prediction for rapidly advancing vaccine and drug development against novel pathogenic targets.

## Author summary

New infectious diseases are emerging at an increasing pace, and the ability to quickly understand how pathogens interact with the human body is more important than ever. One powerful way to gain this understanding is by examining the three-dimensional structures of the proteins involved, like seeing how puzzle pieces fit together. But while thousands of these interactions have been

**Data availability statement:** All data and code used to produce the results here are freely available in this gitlab repository: https://gitlab.com/patrickbryant1/hpopt

**Funding:** This study was supported by the SciLifeLab & Wallenberg Data Driven Life Science Program (grant: KAW 2020.0239, P.B.). Computational resources were enabled by the supercomputing resource Berzelius provided by National Supercomputer Centre at Linköping University and the Knut and Alice Wallenberg foundation with project ids berzelius-2021-29, Berzelius-2023-267, Berzelius-2024-78 and Berzelius-2024-292 (P.B.). M.L. is supported by a Karolinska Institutet faculty-funded Career Position, a Cancerfonden Project grant, the Swedish Research Council (VR) Research Environment Grant, a Consolidator Grant from the Swedish Society for Medical Research (SSMF), and the Knut and Alice Wallenberg foundation (2022.0032). The funders had no role in study design, data collection and analysis, decision to publish, or preparation of the manuscript.

**Competing interests:** The authors have declared that no competing interests exist.

observed experimentally, only a small fraction have known structures. Recent advances in artificial intelligence, such as AlphaFold, now allow us to predict these structures with high accuracy. In this study, we applied structure prediction to over 9000 protein-protein interactions between humans and ten different pathogens. We generated high-confidence structural models for several interactions that previously lacked any structural information. We further investigated one of these predictions, involving the bacterium *Francisella tularensis* and a human immune protein, using native mass spectrometry. Our analysis supports the predicted complex and suggests a potential role in immune system evasion. This work demonstrates how structure prediction can rapidly expand our understanding of host-pathogen interactions and help guide the development of new treatments and vaccines against future infectious threats.

## Introduction

During the recent pandemic outbreak of SARS-COV-2, the importance of obtaining fast insights into an emerging pathogen and its relationship with the host has become clear. Information about the interaction between the Spike protein and the human ACE2 receptor provided essential structural information for vaccine development and design. If this information could have been obtained earlier, it is possible that the pandemic would have had less of an impact on society due to vaccines and treatments being developed and deployed faster.

The interconnectivity of the world is increasing and so is the arms-race between humans and the pathogens we encounter. The same is true for our crops and livestock. Being able to predict host-pathogen protein-protein interactions (HP-PPIs), can yield structural insight and thereby shorten development timelines substantially, generating more profitable and more timely treatments and preventive measures. For the human race to prosper in the coming centuries, we must stay ahead of such evolutionary threats.

The recently developed protein structure prediction methods AlphaFold [1] (AF) and AlphaFold-multimer [2] (AFM) have been shown to greatly outperform other methods in protein complex prediction [3]. Both of these methods rely heavily on multiple sequence alignments (MSAs) [1]. When predicting the structure of protein complexes, these MSAs have to be paired to utilise the evolutionary relationship between orthologous protein sequences. However, none of these methods consider that hosts and pathogens do not have orthologs by definition. As a consequence, it is not known what the performance of AF and AFM is on HP-PPIs compared to predicting interactions within the same species.

Docking proteins with physics-based methods alone, without the use of evolutionary relationships between proteins, has proven to perform poorly [4]. This suggests that it is hard to obtain good results using only physical principles. Recent analyses of protein conformational ranking using AlphaFold [3] suggest that an energy function

far better than any previously known has been learned in AlphaFold. This opens up the possibility to predict highly accurate structures of protein complexes even for HP-PPIs that lack orthologs.

Previously, it has been shown that mammalian hosts and pathogens are under high evolutionary pressure, leading to positive selection [5]. In individual cases, it has been shown that HP-PPIs leave evolutionary marks due to this purifying selection [6,7]. However, apparent positive selection between a host and a pathogen protein does not imply a causal link. The positive selection in the host may be due to a previous pathogen that is now extinct or due to several current pathogens acting together, blurring the so-called "Red Queen" hypothesis [8] of a host-pathogen arms race [5].

Here, we predict the structure of 9452 HP-PPIs between human proteins and ten different pathogens from the HPIDB [9] with AlphaFold [1,4]. We identify novel interfaces and analyse these with native mass spectrometry. The workflow describes how to apply protein structure prediction for the identification of new targets.

## Results

### Structure prediction of known host-pathogen interactions

The FoldDock protocol, based on AlphaFold (AF) and AlphaFold-multimer (AFM) [10], was used to predict the structure of 111 host-pathogen protein-protein interactions (HP-PPIs). In addition, templates were added to FoldDock due to indications that this can improve the accuracy in some cases [11]. The median TM-score from MMalign [12] is 0.64 for Fold-Dock, 0.67 for AFM and 0.68 for FoldDock+templates (1a). However, AFM was trained on all proteins with a release date earlier than 2018-04-30. This leaves only 24 out of 111 structures (22%) to test this method, and the median TM-score for AFM is reduced to 0.63, 0.67 for FoldDock+templates and increased to 0.65 for FoldDock on this set (Fig 1b).

The pDockQ score from FoldDock has been shown to discriminate true PPIs and their structural accuracy. To test the structural quality correspondence for HP-PPIs, we calculate pDockQ for the 111 HP-PPIs using FoldDock (Fig 1c) and FoldDock+templates (Fig A in S1 Appendix). We find that the TM-score increases with pDockQ for FoldDock, and a cutoff of 0.3 in pDockQ roughly corresponds to an average TM-score of 0.9 and above. Using FoldDock+templates, however, results in reduced resolution between good and bad models and thereby more false positives.

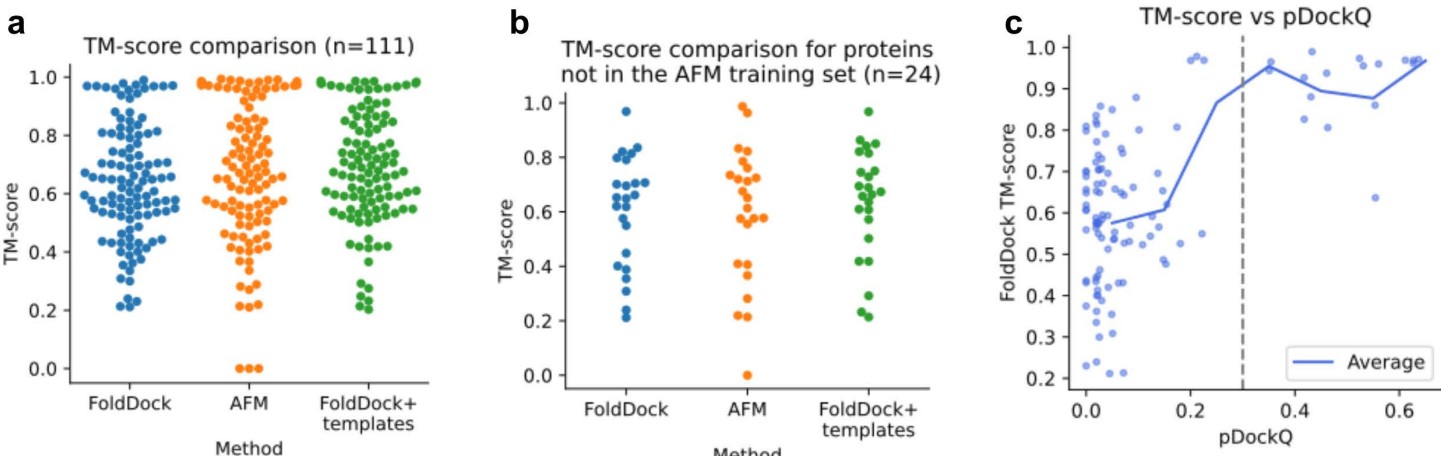

**Fig 1. TM-score comparison between FoldDock and AFM. a) The median TM-score is 0.67 for AFM, 0.64 for FoldDock and 0.68 for FoldDock+-templates, using all 111 nonredundant host-pathogen interactions in the PDB. b) Selecting only the host-pathogen interactions that are not present in the AFM training set (n = 24), reduced the median TM-score from AFM to 0.63, to 0.67 for FoldDock+templates and increased it for FoldDock to 0.65. c) Comparison of pDockQ and the TM-score using FoldDock (n = 111).** The points represent each model, the solid blue line the running average using a step size of 0.1 in pDockQ and the dashed grey line a cutoff of 0.3 in pDockQ. When the pDockQ score is high, so is the TM-score.

Constructing an ROC curve using pDockQ as a separator, with positive examples here having a TM-score over 0.9, 87% of the models above the TM-score 0.9 can be called correct at a FPR of 5% using FoldDock without templates (ROC AUC = 0.97 for FoldDock and 0.95 for FoldDock+templates, Fig A in S1 Appendix). The reason for using the TM-score instead of the DockQ [13] score is because the full genetic sequences were used here, meaning that the predicted and native structures have different lengths, something DockQ handles poorly (see Fig B in S1 Appendix). Due to the better discriminatory ability using FoldDock alone, we abandon the other methods and continue only with the FoldDock protocol in all subsequent analyses.

## Structure prediction of novel host-pathogen interactions

The host-pathogen interaction database [9] 3.0 contains 69'787 curated interactions in total, of which 32'458 are within humans, and 21'064 of these have direct interactions or physical associations. We selected the top ten unique pathogens in terms of interaction prevalence (Fig 2a); Dengue virus type 2 (Dengue), Human T-cell leukemia virus 1 (HTLV-1), Influenza A virus, *Coxiella burnetii*, Epstein-Barr virus (EBV), Human papillomavirus type 16 (HPV), Hepatitis C virus genotype 1b (HCV), *Francisella tularensis* subsp. *tularensis* (*F. tularensis)*, *Bacillus anthracis* and *Yersinia pestis*. In total, there are 9576 interactions (45% of all human interactions) with 4051 unique human proteins and 2560 unique pathogenic proteins

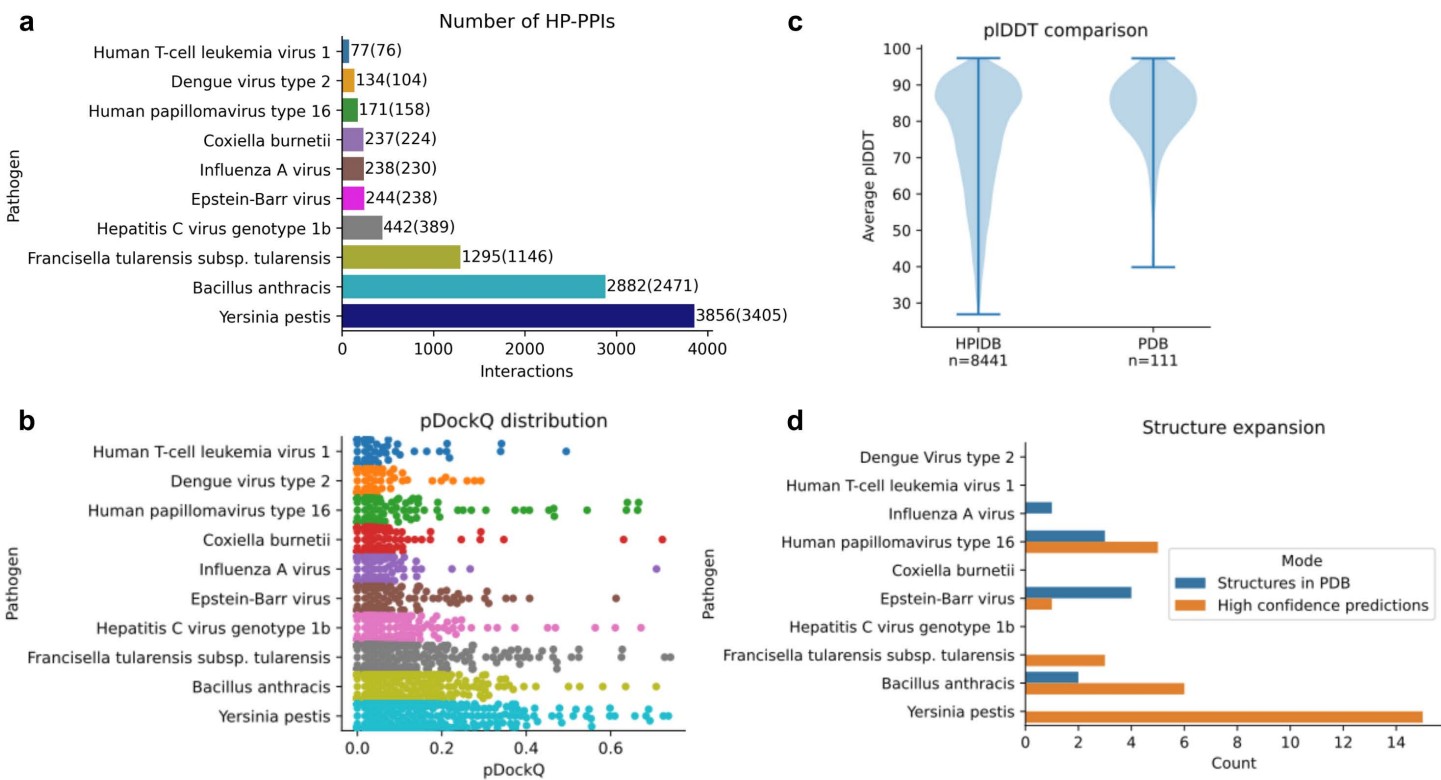

**Fig 2. Structure prediction for the 10 selected pathogens. a)** Number of interactions with human proteins for each pathogen in the HPIDB. The numbers in parentheses are the number of interactions that could be predicted (some proteins are too large to be predicted in complex on GPUs with 40Gb RAM, limit of approximately 3000 residues). **b)** Distribution of pDockQ scores for the 8441 HP-PPIs that were successfully predicted. **c)** Distribution of the average pIDDT for each chain in the complexes from the HPIDB (n = 8441) and the PDB (n = 111). The PDB set has much higher pIDDT on average. There are 93 complexes out of 111 where both chains have over 70 pIDDT on average for the PDB set (84%) and 3472 out of 7948 for the HPIDB set (44%). **d)** The number of HP-PPIs with known complex structure (n = 10) and with high-quality predictions (above 70 pIDDT and pDockQ 0.3, n = **30**) for each pathogen. Five of these are from HPV, one from EBV, three from *F. tularensis*, six from *B. anthracis* and 15 from *Y. pestis*.

among these interactions. In total, 8441 HP-PPIs (88%) were successfully predicted. Fig 2b shows the DockQ distributions for each pathogen. Only a small fraction of the total number of HP-PPIs report a pDockQ score above 0.3. This finding is consistent with the evaluation of the known HP-PPIs from PDB (Fig 1c and Table A in S1 Appendix for summary statistics of all predictions and scores).

The structures in PDB will be highly ordered, even considering the full-length sequences. This is because disordered proteins are hard to crystallise and thereby, infrequent in structural databases. In reality, many proteins are highly flexible, and many viruses have highly disordered or non-structural proteins [14]. To account for this bias, we analyse the protein disorder distribution measured by the predicted lDDT (plDDT) from AlphaFold using the HP-PPIs from PDB and compare it to the set from the HPIDB (Fig 2c). As expected, the PDB set has higher plDDT on average. To account for this discrepancy, we select only the complexes where both chains have an average plDDT ≥ 70 (43% of the HPIDB complex predictions and 84% of the PDB set) and do not contain any clashes between interacting residues (CBs within 1Å from each other). In some cases, one chain is predicted to intersect the other, although there are no clashes (Fig C in S1 Appendix). These cases were removed as well.

Fig 2d shows the number of HP-PPIs with known complex structure (n = 10) and the number of high-quality predictions with an expected TM-score of 0.9 (above 70 plDDT and pDockQ 0.3, n = 30). None of the interactions with known structures were accurately predicted. In total, there are 30 new predictions, expanding the number of available structures fourfold. *F. tularensis subsp. tularensis* and *Y. pestis*, which have no known HP-PPI structures, here report 3 and 15 high-quality predictions respectively. *B. anthracis*, Epstein-Barr virus and Human papillomavirus type 16 have 6, 1 and 5 predictions, respectively, compared to 2, 4 and 3 known structures.

## Identification of structural targets

The prediction and identification of high-quality HP-PPI structures enable the extraction of key parts of these complexes. To illustrate the principle of AI-first target identification, all 30 high-quality predictions were analysed. Fig 3 displays 9 of these predictions with the highest pDockQ scores for different organisms. One interaction is from HPV (a), one from EBV (b), two from *F. tularensis* (c&d), two from *B. anthracis* (e&f) and three from *Y. pestis* (g,h &i) (see S1 Appendix "High quality targets from the HPIDB" for an analysis of all 30 predictions).

The predicted structure of the interaction between genes UBA1 (P22314, human) and E2 (P03120, HPV) is displayed in Fig 3a. UBA1 is an enzyme involved in the catalysis of the first step in ubiquitin conjugation to mark cellular proteins for degradation [15]. E2, which is involved in viral DNA replication [16], has been found to regulate ubiquitination by the host and thereby hinder degradation by the proteasome [17]. The position occupied by E2 traps ubiquitin in the native structure. This suggests that E2 binds to UBA1 and hinders the activation of ubiquitin sterically.

The only high-quality prediction for EBC is between TTC12 (https://www.uniprot.org/uniprot/Q9H892) and BBRF2 (https://www.uniprot.org/uniprot/P29882), Fig 3b. TTC12 is a protein that plays a role in the assembly of motile cilia and is found in the cytoplasm (no available structure in the PDB). BBRF2 is critical for virus egress and is part of the final step in envelope acquisition from the host cytoplasm (closest structure: https://www.rcsb.org/3d-view/6LQN/1). A possible mode of action for viral egress is that TTC12 helps to assemble the capsid, thereby facilitating viral egress through interactions with BBRF2 (it looks like TTC12 grabs hold of BBRF2).

Two predictions for *F. tularensis* are shown in Fig 3c and d. IGKC (membrane-bound immunoglobulin, https://www.uniprot.org/uniprot/P01834) interacts with IPD (https://www.uniprot.org/uniprot/Q5NEX4) in c. IPD may inhibit the IGKC through this interaction, preventing immune response towards other pathogenic particles. LNPEP (Leucyl-cystinyl aminopeptidase, https://www.uniprot.org/uniprot/Q9UIQ6) interacts with argS (Arginine-tRNA ligase, https://www.uniprot.org/uniprot/Q5NHI8) in d. This interaction may inhibit the activity of LNPEP sterically, although the substrate binding site is not hindered.

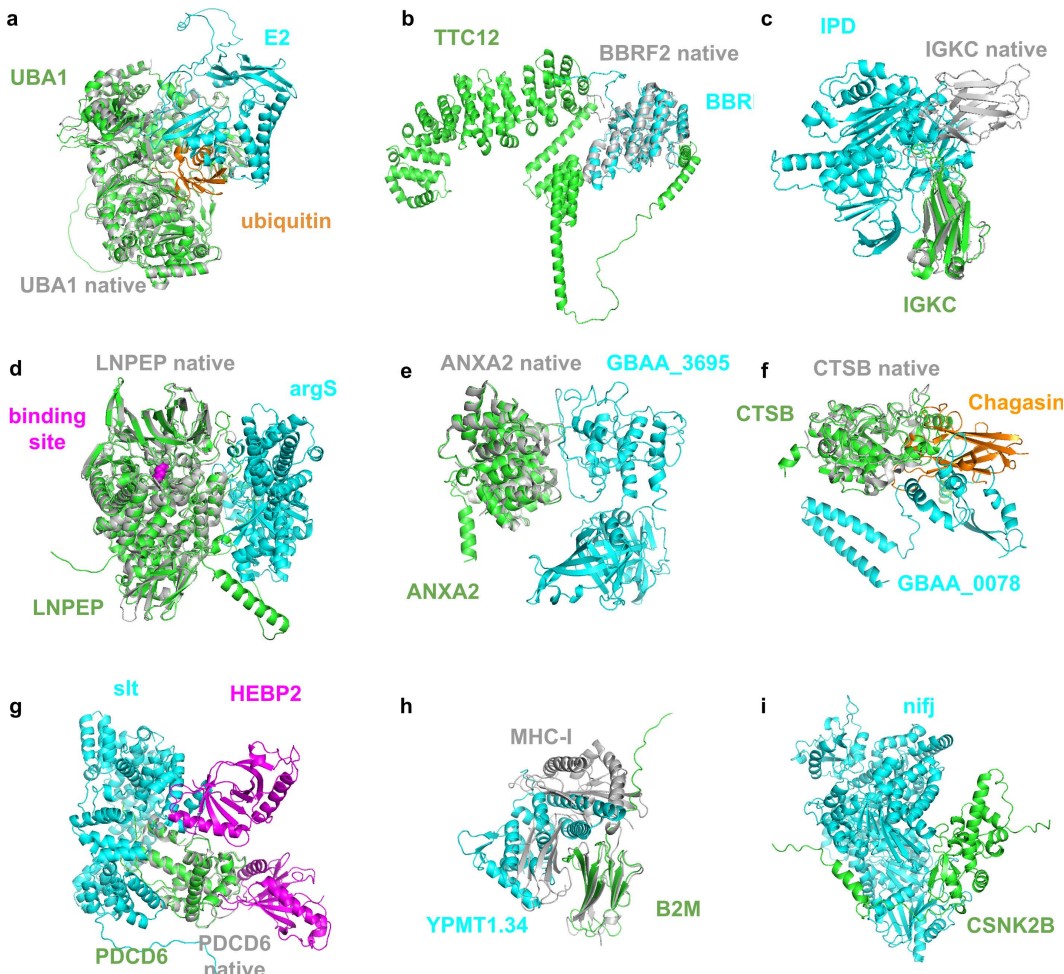

**Fig 3. Analysis of the selected high-quality HP-PPI for HPV (a), EBV (b), *F. tularensis* (c & d), *B. anthracis* (e & f) and *Y. pestis* (g,h &i).** The human proteins are shown in green and the pathogenic ones in cyan. Potential native structures are shown in grey superposed with the predictions. **a)** Predicted structure of the interaction between UBA1 (green) and E2 (cyan. UBA1 is in structural superposition with the native structure (grey, TM-score = 0.98) from HPV. The native structure of UBA1 is complex with ubiquitin (orange, PDB ID 6DC6, https://www.rcsb.org/structure/6dc6). E2 captures ubiquitin in its activation area and thereby likely prevents its release. **b)** Predicted structure of the interaction between TTC12 (green) and BBRF2 (cyan) from EBV. BBRF2 is in structural superposition with the native structure (grey, https://www.rcsb.org/3d-view/6LQN/1). **c)** IGKC (https://www.uniprot.org/uniprot/P01834) interacts with IPD (https://www.uniprot.org/uniprot/Q5NEX4), potentially hindering antibody formation. **d)** LNPEP (https://www.uniprot.org/uniprot/Q9UIQ6) interacts with argS (https://www.uniprot.org/uniprot/Q5NHI8). This interaction may inhibit the activity of LNPEP sterically, although the substrate binding site (magenta) is not hindered. **e)** ANXA2 (Annexin A2, https://www.uniprot.org/uniprot/P07355) interacting with GBAA_3695 (Unknown protein, https://www.uniprot.org/uniprot/A0A0F7RE19). This interaction may inhibit the production of reactive oxygen species. **f)** CTSB (Cathepsin B, https://www.uniprot.org/uniprot/P07858) interacts with GBAA_0078 (UVR domain-containing protein, https://www.uniprot.org/uniprot/A0A6L8P7D1). The structure of Chagasin (https://www.rcsb.org/structure/3CBJ) is shown as it is interacting with CTSB. **g)** PDCD6 (Programmed cell death protein 6, https://www.uniprot.org/uniprot/O75340) interacts with slt (Peptidoglycan lytic exotransglycosylase, https://www.uniprot.org/uniprot/Q8CZP1). HEBP2 is also shown bound to PDCD6 (https://www.rcsb.org/3d-view/5GQQ), which is thought to promote the inhibition of HIV production. **h)** B2M (Beta-2-microglobulin, https://www.uniprot.org/uniprot/P61769, superposition with the structure of MHC-I: https://www.rcsb.org/structure/1A1M) interacts with YPMT1.34 (Uncharacterized, https://www.uniprot.org/uniprot/O68752) **i)** CSNK2B (Casein kinase II subunit beta, https://www.uniprot.org/uniprot/P67870) and nifj (Putative pyruvate-flavodoxin oxidoreductase, https://www.uniprot.org/uniprot/A0A3N4BEU0) interact in Fig H in S1 Appendix.

In Fig 3e and 3f, one can see two predictions from *B. anthracis.* In e, ANXA2 (Annexin A2, https://www.uniprot.org/uniprot/P07355) interacts with GBAA_3695 (Unknown protein, https://www.uniprot.org/uniprot/A0A0F7RE19). This interaction may inhibit the production of reactive oxygen species [18]. In f, CTSB (Cathepsin B, https://www.uniprot.org/uniprot/P07858) interacts with GBAA_0078 (UVR domain-containing protein, https://www.uniprot.org/uniprot/A0A6L8P7D1). At the same interaction site, CTSB is inhibited by binding to the protein Chagasin from Trypanosoma cruzi [19], preventing protein degradation.

Finally, three predictions from *Y. pestis* are shown in Fig 3g-i. In g, PDCD6 (Programmed cell death protein 6, https://www.uniprot.org/uniprot/O75340) interacts with slt (Peptidoglycan lytic exotransglycosylase, https://www.uniprot.org/uniprot/Q8CZP1). HEBP2 is also shown bound to PDCD6 (https://www.rcsb.org/3d-view/5GQQ), which is thought to promote the inhibition of HIV production [20]. The interaction between slt and PDCD6 may work similarly. B2M (Beta-2-microglobulin, https://www.uniprot.org/uniprot/P61769, superposition with the structure of MHC-I: https://www.rcsb.org/structure/1A1M) interacts with YPMT1.34 (Uncharacterized, https://www.uniprot.org/uniprot/O68752) in h, which may have an inhibitory effect on MHC-I formation and, thereby, antigen presentation. CSNK2B(Casein kinase II subunit beta, https://www.uniprot.org/uniprot/P67870) and nifj (Putative pyruvate-flavodoxin oxidoreductase, https://www.uniprot.org/uniprot/A0A3N4BEU0) interact in i. CSNK2B participates in Wnt signalling, which is important for immune surveillance, which means that this may be avoided or down-regulated through the interaction with nifj.

## Mass spectrometric validation of the IPD-IGKC complex

As outlined above, all of the 9 top-scoring complexes were found to contain at least one protein without experimentally determined structure. We selected the complex in Fig 3c between IPD and IGKC since a potential interaction has previously been identified in a yeast two-hybrid screen [21]. The high-resolution structure of IPD from *Neisseria meningitidis,* which shares 59% sequence identity with IPD from Francisella tularensis, has been determined by X-ray crystallography (PDB ID 1OJT). The superposition of the structures reveals a highly conserved fold, except for a small β-hairpin covering residues 246–257 of Neisseria IPD (Fig 4a). Importantly, Neisseria IPD is a constitutive dimer. To generate the heterotetramer (Fig 4b), two of the predicted IPD-IGKC heterodimers were combined into a heterotetramer by aligning each IPD protein in the heterodimer with a subunit of the native IPD dimer structure (PDB ID 1OJT). The model suggests that bound IGKC faces away from the dimer interface. If the prediction is correct, dimeric Francisella IPD should bind two IGKC monomers (Fig 4b).

To test this hypothesis, we recombinantly produced both proteins in E. coli and analyzed the interaction using native mass spectrometry (nMS), an orthogonal method for the validation of machine learning-based structure prediction [22]. Briefly, nMS relies on the gentle transfer of intact protein complexes from solution to the gas phase, enabling direct determination of their native stoichiometries. nMS analysis of IPD shows exclusively dimeric protein, in agreement with the homologous Neisseria IPD structure (Fig 4c). IGKC is mostly monomeric in nMS, with a minor population (< 25% of the total protein) being dimeric (Fig 4c). Incubation and subsequent nMS analysis of equimolar amounts of IPD and IGKC results in the formation of additional peaks in the spectrum, which correspond in mass to a hetero-tetrameric complex composed of two IPD and two IGKC subunits (predicted mass: 130 kDa, measured mass: 130.3 kDa). We also detected minor peaks corresponding to a complex of two IPD and one IGKC protein (predicted mass: 117.3 kDa, measured mass: 117.5 kDa). Simultaneously, the oligomeric state of IGKC is shifted towards dimers, which indicates depletion of the monomeric species (Fig 4c). From these observations, we conclude that the most likely explanation is that the IPD dimer binds two IGKC monomers in a 1:2:1 architecture, in agreement with the AI model. We note that a 2:2 complex of one IPD and one IGKC dimer each cannot be excluded based on the MS data, but appears highly unlikely.

## Discussion

With the advent of highly accurate structure prediction, exemplified by AlphaFold2, it has become possible to systematically expand structural knowledge across a wide range of organisms [23]. This technological leap opens entirely new

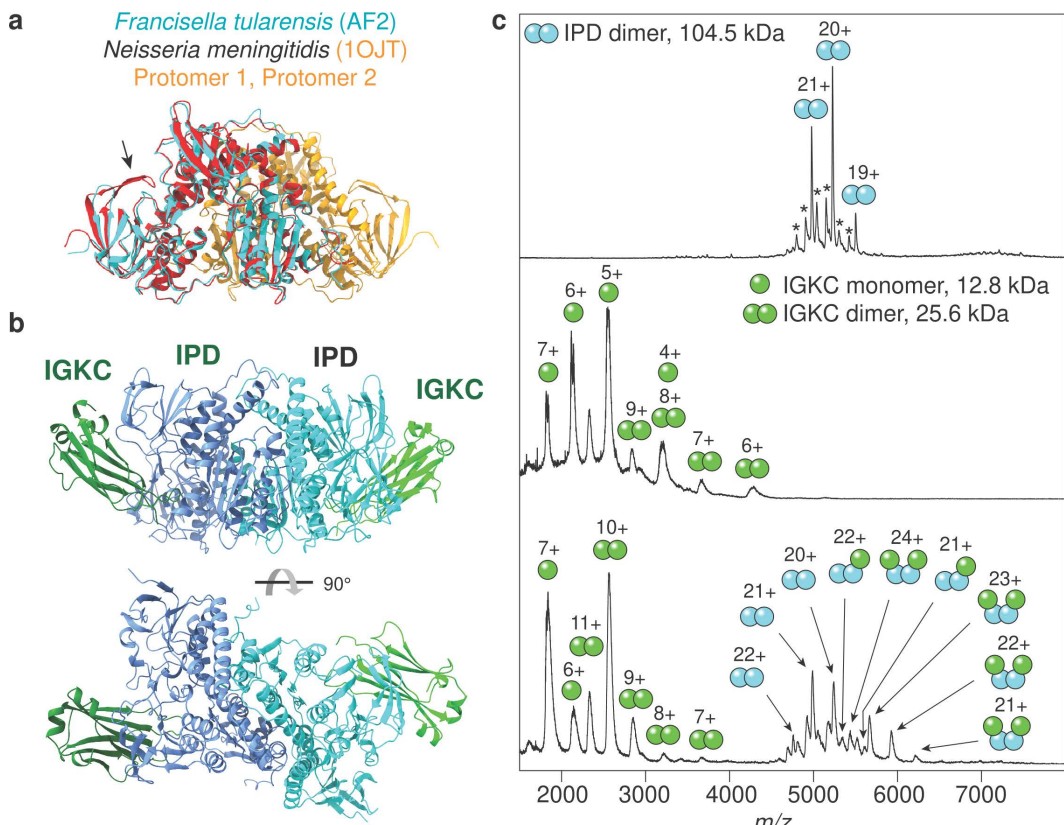

**Fig 4. Mass spectrometric validation of the IPD-IGKC interaction. a)** Overlay of the predicted structure of *F. tularensis* IPD (cyan) with the crystal structure of the IPD dimer from *N. meningitidis* (red/orange). The structures are nearly identical except for a β-hairpin (arrow). **b)** Two of the predicted IPD-IGKC heterodimers were combined into a heterotetramer by aligning each IPD protein in the heterodimer with a subunit of the native IPD dimer structure from Fig 4a (PDB ID 1OJT). Accordingly, each IPD protomer can bind one IGKC monomer. **c)** Native Mass Spectrometry analysis of *Francisella* IPD (top) shows exclusively dimeric protein. Minor peaks indicated by asterisks correspond to homo-and heterodimers involving a truncated variant (103.1 kDa). Human IGKC (middle) exists predominantly as a monomer with a smaller dimeric population. Incubation of equimolar amounts of IPD and IGKC (bottom) results in the formation of a 2:2 IPD-IGKC complex, a minor population of 2:1 IPD-IGKC complex, and a shift in the oligomeric state for IGKC towards the dimer.

prospects for rational vaccine and drug development by enabling rapid identification of potential therapeutic targets. In this study, we present an AI-guided framework for host-pathogen structure prediction, aimed at uncovering novel interactions of functional and clinical relevance.

Among the ten studied organisms, 30 predicted protein pairs have an expected TM-score of 0.9. We focused on a high-confidence interaction between a bacterial dihydrolipoyl dehydrogenase and the human immunoglobulin kappa constant (IGKC), a key component of the humoral immune system. We investigated this complex using native mass spectrometry, which supported the predicted assembly. Notably, clinical studies have shown that patients with impaired IGKC synthesis are highly susceptible to bacterial infections, highlighting the potential advantage for pathogens in targeting or sequestering this molecule [24].

These findings demonstrate the potential of our AI-first screening approach to reveal novel interfaces in host-pathogen interactions that may play a role in immune evasion and that represent promising candidates for future in vivo validation and therapeutic exploration.

## Methods

### Nonredundant host-pathogen complexes from the PDB

All heteromeric protein structures with below 5 Å resolution and experimental technique X-ray crystallography or electron microscopy were downloaded from the PDB on 2021-12-20. From these structures, PFAM domains and species were mapped to Uniprot KB [25], keeping all structures with UniprotKB annotations. All structures that contain interacting sequences from at least two different Superkingdoms and have different PFAM domains were thereafter selected. The structures with unique pairwise interacting PFAM domains containing the most contacts and oldest release date were chosen (n = 111, 52 of these include human proteins). A contact is defined as two different chains having beta carbons within 8 Å from each other (alpha carbons for glycine). Fig 5 provides a visual guide to the data selection workflow.

In total, there are 134 HP PPIs with uniquely interacting PFAM domains. Of these interactions, 24 are immune reactions and were therefore excluded as the objective here is to predict specific HP interactions and not general ones (in theory there exist antibodies that can bind any proteins). The final dataset consists of 111 HP interactions originating from 211 unique protein chains. The hosts mainly consist of Eukarya, with *Homo sapiens* being the most represented species, while the pathogens are mainly Bacteria and Viruses (Fig 6). It is evident that the host distribution is severely biassed towards the most studied eukaryotic (*H. sapiens, M. musculus and S. cerevisiae*) and Bacterial species (*E. coli*).

### Human-pathogen PPIs from the HPIDB

The host-pathogen interaction database [9] v 3.0 contains 69'787 curated interactions in total, of which 32'458 are within humans, and 21'064 of these have direct interactions or physical associations. We selected the top ten unique pathogens in terms of interaction prevalence; Dengue virus type 2, Human T-cell leukemia virus 1, Influenza A virus, *Coxiella burnetii*, Epstein-Barr virus, Human papillomavirus type 16, Hepatitis C virus genotype 1b, *Francisella tularensis subsp. tularensis, Bacillus anthracis* and *Yersinia pestis*. In total, there are 10563 interactions (50% of all human interactions, Fig 2a).

There are 4153 unique human proteins and 2663 pathogenic proteins among these interactions. For the human proteins, 4130 out of 4153 UniProtKB AC/ID identifiers were successfully mapped to 4110 unique UniProtKB IDs with available sequences. For the pathogenic proteins, 2645 out of 2646 UniProtKB AC/ID identifiers were successfully mapped

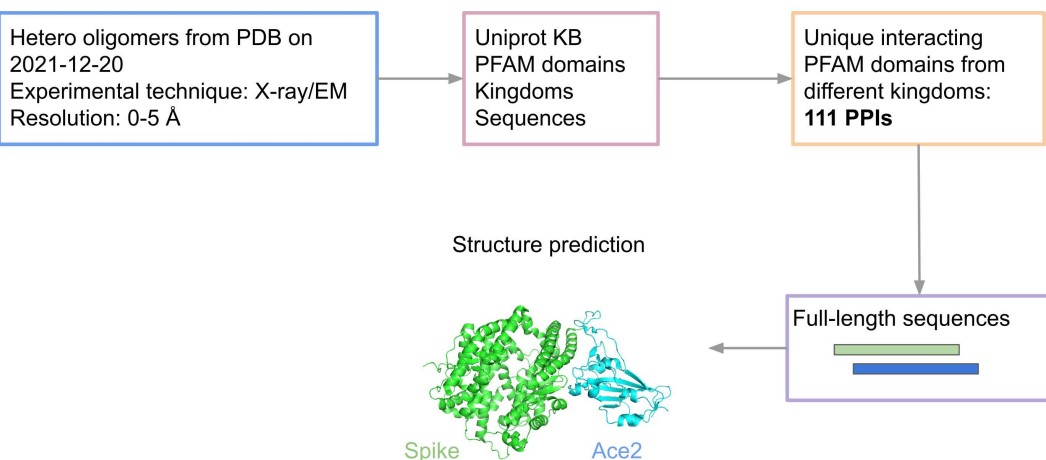

**Fig 5. Data and modelling workflow.** To extract host-pathogen protein complexes from the PDB we selected all heteromers with a resolution below 5Å. We then mapped the proteins to PFAM and selected unique combinations of interacting domains to exclude redundancy in the interactions. The structure of the resulting 111 host-pathogen interactions were then predicted using the full-length sequences.

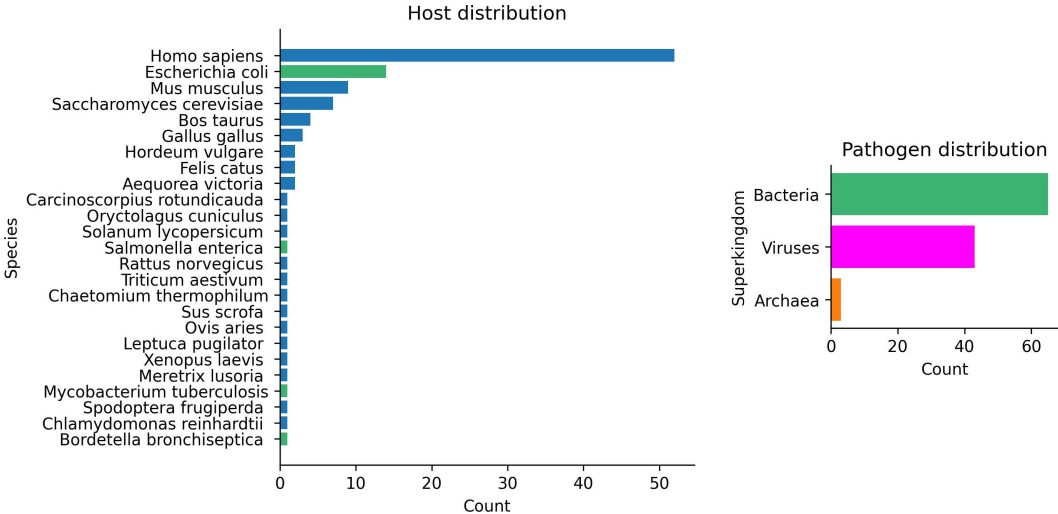

**Fig 6. Host (species) and pathogen (superkingdoms) distributions for the 111 PPIs. Eukarya** are shown in blue, **Bacteria** in green, **Viruses** in magenta and **Archaea** in orange. The majority of hosts are Eukarya, with humans being the most represented species. Most pathogens come from Bacteria, followed by Viruses. The host distribution is severely biased towards the most studied eukaryotic (*Homo sapiens, Mus musculus and S. cerevisiae*) and Bacterial species (*E. coli*).

to 2563 unique UniProtKB IDs with sequence. After the mapping to UniProtKB, 9452 out of 10563 interactions (89.5%) remain in total, including 4026 unique human proteins and 2545 pathogenic proteins among these interactions.

## Poly-protein product extraction

Dengue virus type 2 and Hepatitis C virus genotype 1b contain large poly-proteins which produce several protein products as they are processed. In the HPIDB, the exact interactions between different poly-protein products and human proteins are not listed. Therefore, the original publications reporting these interactions were analysed. For the Dengue virus, 187 interactions were taken from the experimental study [26], annotated in the HPIDB, and the poly-protein products were extracted using the uniprot annotations from the uniprot ID P29991. After mapping the corresponding human genes to uniprot to retrieve sequences, 134 human-dengue interactions remain. For Hepatitis C, 481 interactions were taken from the experimental study [27] annotated in the HPIDB, and the poly-protein products were extracted using the uniprot annotations from the uniprot ID P26664. The F protein produced by an overlapping reading frame was taken from uniprot ID P0C045. After mapping the corresponding human genes to uniprot to retrieve sequences, 442 human-hepatitis C interactions remain.

## FoldDock pipeline

We modelled complexes using AlphaFold2 (AF2) with a modified script to introduce a 200-residue chain break [28]. We used MSAs generated by searching uniclust30_2018_08[32] with HHblits (from HH-suite[33] version 3.1.0) with the options:

```
hhblits -E 0.001 -all -oa3m -n 2
```

The resulting MSAs are block diagonalized as pairing should result in zero matches (the host and pathogen should not have common ancestral proteins).

## FoldDock with templates

Templates were added to FoldDock due to indications that this can improve the accuracy in some cases [11]. The default template methodology used in AlphaFold was used. First, Uniref90 [11, 29] is searched using hhsearch [30] (from hh-suite

v.3.0-beta.3 version 14/07/2017). The resulting MSA is then parsed and used to search PDB70 with hhsearch [30] (from hh-suite v.3.0-beta.3 version 14/07/2017). The hits from PDB70 are extracted and used as input to AlphaFold through its feature-processing pipeline. To ensure the actual structure being predicted is not identical to the ones used as templates, AF checks the provided PDB code as well as the sequence similarity towards the templates. If the sequence similarity is above 95%, the template will not be used.

One can provide templates in any order, as the template sequence is aligned to the query and the atom positions are matched. In total, 20 templates were used, which were later filtered to the four best-matching ones by the AlphaFold data processing pipeline.

## AlphaFold-multimer

AlphaFold-multimer (AFM) [10] was trained on all data used here with a release date earlier than 2018-04-30. This leaves only 24 out of 111 structures (22%) to test this method. This is, however, not of importance, as the median DockQ score is 0.01 using all 111 structures (Fig 1). AFM creates four different MSAs for each single protein chain. Three of the MSAs are generated using jackhmmer from HMMER3 [31] and searching Uniref90 v.2020_01 [32], Uniprot v.2021_04 [25] and MGnify v.2018_12 [33] and the fourth with HHBlits [30] (from hh-suite v.3.0-beta.3 version 14/07/2017) by searching the Big Fantastic Database [34] (BFD) and uniclust30_2018_08 [35]

The Uniprot search results are used for MSA pairing, assuming orthologous relationships are present. The remaining MSAs are used for block-diagonalization. Both the paired and block-diagonalized MSAs are then used to predict the structure of protein complexes. It is currently not possible to supply optimised MSAs to AFM (as it is to AF2) due to the MSA pairing and feature processing being made within the feature processing pipeline.

## Scoring with MMalign

The program MMalign [12] was used to score the predictions of host-pathogen complexes extracted from the PDB. This provides a complementary assessment to the DockQ score [13] which the confidence metric pDockQ used to select true accurate predictions is fitted to.

pDockQ

$$pDockQ = \frac{0.724}{1 + e^{-0.052(x-152.611)}} + 0.018$$

(1)

Where x is the average interface plDDT multiplied with the logarithm of the number of interface contacts (CBs within 8 Å).

## Selecting high-quality predictions

Out of the 8441 human-pathogen interactions from the HPIDB that could be predicted, there are 3593 complexes where both chains have above 70 plDDT (42.57%), compared with 93 complexes out of 111 above 70 plDDT for the set of known structures from the PDB (83.78%). Out of the 3593, 52 complexes have a pDockQ score above 0.3 and **30** of these are without clashes.

## Protein preparation

The genes for MGH6-IGKC and MGH6-IPD in the pET26b(+) expression vector were transformed into chemically competent *E. coli* BL21 (DE3) cells. Overnight cultures were inoculated 1:100 to Luria-Bertani medium containing 50 µg/mL kanamycin. The cultures were grown at +37°C to an OD600 of 0.6-0.9 before induction with 0.5 mM isopropyl-β-D-thiogalactopyranoside (IPTG) and overnight expression at +25°C. For each construct, 250 mL of culture was harvested by centrifugation at 6000×g for 20 min at RT. The pellet was resuspended in 20 mM Tris, pH 8.0 buffer (1 mL for 10 mL of

culture) and stored at -20°C overnight. After thawing, the cells were lysed with the addition of 1 mg/mL lysozyme, 10 µg/ml DNAse I and 2 mM MgCl2 and incubated for 1 hour on ice followed by sonication using a probe sonicator (Qsonica, CT) on ice for 5 min, 0.5 s on/1 s off at 30% power. The samples were centrifugated for 15 min at 15 000 × g, 4°C, the supernatant was decanted, and the pellet was resuspended in 20 mM Tris, 500 mM NaCl pH 8 buffer (1 mL per 10 mL culture) sonicated and centrifuged as before. Two further resuspensions with the same parameters were conducted. Samples taken from steps during expression, lysis and purification were analyzed using SDS-PAGE 4–20% Mini-PROTEAN TGX polyacrylamide gels (Bio-rad Laboratories Inc., USA) stained with Coomassie G-250 Brilliant Blue dye. The protein-containing supernatants were supplemented with 20 mM imidazole and 500 mM NaCl if needed, loaded onto 1 mL HiTrap IMAC HP (Cytiva, Sweden) columns charged with Ni2+ and eluted with a linear gradient of buffer containing 500 mM imidazole, 500 mM NaCl, 20 mM Tris, pH 8. Fractions were analysed using SDS-PAGE gels, protein-containing fractions were pooled, concentrated using appropriate MWCO Amicon ® Ultra 15 centrifugation tubes (Merck, USA) and dialysed at +4°C five times against 20 mM Tris, 500 mM NaCl, pH 8 using Slide-A-Lyzer™ MINI Dialysis Devices with 10K MWCO (Thermo Scientific, USA). Protein concentrations were determined using Nanodrop (Thermo Scientific, USA) measurements with the specific extinction coefficients and molecular weights.

## Mass spectrometry

Purified IPD and IGKC at a concentration of 50 µM were buffer-exchanged into 200 mM ammonium acetate, pH 6.9, using Bio-Spin P-6 columns (BioRad, CA). For complex formation, both proteins were incubated at pH 5.5 for 30 min at 37 °C prior to analysis. Mass spectra were acquired on a Waters Synapt G1 travelling wave ion mobility mass spectrometer modified for high-mass analysis (MS Vision, NL) equipped with an offline nanospray source. The capillary voltage was 1.5 kV, the source pressure was 8 mbar, and the source temperature was 80°C. Mass spectra were visualized using Mass-Lynx 4.1 (Waters, UK).

## Supporting information

**S1 Appendix.  Fig A.** a) Comparison of pDockQ and the TM-score using FoldDock (n = 111) and FoldDock+templates. The points represent each individual model, the solid lines the running averages using a step size of 0.1 in pDockQ and the dashed grey line a cutoff of 0.3 in pDockQ. When the pDockQ score is high, so is the TM-score. There are more models with low TM-scores at high pDockQ scores using FoldDock+templates. b) ROC curve using pDockQ as a separator for the 111 HP-PPIs with known structure for the standard FoldDock approach (std) and using FoldDock+templates (templates). Positive examples here have a TM-score over 0.9. At an FPR of 5%, 87% of the TP models can be called correct using the std FoldDock model. **Fig B.** An example of an accurate prediction (6OAM_A-6OAM_d), where DockQ reports a low score (DockQ = 0.007) and MMalign a high score (TM-score = 0.97). This exemplifies the need to take the length difference between native and predicted structures into account and supports the use of MMalign. **Fig C.** Example of an interaction where one chain is predicted to intersect the other, although there are no clashes (Q14318-Q8D097). These cases were removed as well. **Fig D.** Analysis of the high quality HP-PPIs for HPV. The human proteins are shown in green and the pathogenic ones in cyan. Potential native structures are shown in grey superposed with the predictions. a) Predicted structure of the interaction between UBA1 (https://www.uniprot.org/uniprot/P22314) and E2 (https://www.uniprot.org/uniprot/P03120) with the interface residues coloured in magenta. UBA1 is in structural superposition with the native structure (grey, TM-score = 0.98). The native structure of UBA1 is complex with ubiquitin (PDB ID 6DC6, https://www.rcsb.org/structure/6dc6) and with E2 superposed according to the predicted structure. E2 captures ubiquitin in its activation area and thereby likely prevents its release. b) Interaction between SRP19 (https://www.uniprot.org/uniprot/P09132), crucial for ribosome binding, and HPV protein E7 (https://www.uniprot.org/uniprot/P03129) which is involved in regulatory mechanisms such as transcriptional activation. In its native form, SRP19 interacts with RNA (https://www.rcsb.org/structure/1JID)

c-e) Interactions between various forms of human proteins which facilitate nuclear import of other proteins (TNPO1 https://www.uniprot.org/uniprot/Q92973, KPNB1 https://www.uniprot.org/uniprot/Q14974 and IPO5 https://www.uniprot.org/uniprot/O00410) and viral proteins E6 (an oncoprotein that destroys many host cell regulatory proteins, https://www.uniprot.org/uniprot/P03126) and L1 (the viral capsid protein, https://www.uniprot.org/uniprot/P03101). **Fig E.** Analysis of the high-quality HP-PPI for EBV. The human proteins are shown in green and the pathogenic ones in cyan. Potential native structures are shown in grey superposed with the predictions. Predicted structure of the interaction between TTC12 (https://www.uniprot.org/uniprot/Q9H892) and BBRF2 (https://www.uniprot.org/uniprot/P29882). BBRF2 is in structural superposition with the native structure (grey, https://www.rcsb.org/3d-view/6LQN/1). **Fig F.** Analysis of the high quality HP-PPI for F. tularensis. The human proteins are shown in green and the pathogenic ones in cyan. Potential native structures are shown in grey superposed with the predictions. a) IGKC (https://www.uniprot.org/uniprot/P01834) interacts with IPD (https://www.uniprot.org/uniprot/Q5NEX4), potentially hindering antibody formation. b) TRPC1 (https://www.uniprot.org/uniprot/P48995) interacts with purL (https://www.uniprot.org/uniprot/Q5NEC0). No clear biological mechanism can be deduced from this interaction. c) LNPEP (https://www.uniprot.org/uniprot/Q9UIQ6) interacts with argS (https://www.uniprot.org/uniprot/Q5NHI8). This interaction may inhibit the activity of LNPEP sterically, although the substrate binding site (magenta) is not hindered. **Fig G.** Analysis of the high quality HP-PPI for B. anthracis. The human proteins are shown in green and the pathogenic ones in cyan. Potential native structures are shown in grey superposed with the predictions. a) XPO1 (Exportin-1, https://www.uniprot.org/uniprot/O14980) interacts with dacB (Diadenylate cyclase, https://www.uniprot.org/uniprot/A0A6L8PQ13) b) PFAS (Phosphoribosylformylglycinamidine synthase, https://www.uniprot.org/uniprot/O15067) interacts with leuS (Leucine-tRNA ligase, https://www.uniprot.org/uniprot/Q81KK6) c) ANXA2 (Annexin A2, https://www.uniprot.org/uniprot/P07355) interacting with GBAA_3695 (Unknown protein, https://www.uniprot.org/uniprot/A0A0F7RE19) d) LMNA (Prelamin A/C, https://www.uniprot.org/uniprot/P02545) interacts with GBAA_0983 (putative membrane protein, https://www.uniprot.org/uniprot/A0A6L8P438) e) CTSB (Cathepsin B, https://www.uniprot.org/uniprot/P07858) interacts with GBAA_0078 (UVR domain-containing protein, https://www.uniprot.org/uniprot/A0A6L8P7D1). The structure of Chagasin (https://www.rcsb.org/structure/3CBJ) is shown as it is interacting with CTSB. f) WASHC5 (WASH complex subunit 5, https://www.uniprot.org/uniprot/Q12768) and vpR (Minor extracellular protease VpR, https://www.uniprot.org/uniprot/A0A6H3AKS4). **Fig H.** Analysis of the high quality HP-PPI for B. anthracis. The human proteins are shown in green and the pathogenic ones in cyan. Potential native structures are shown in grey superposed with the predictions. a) PDCD6 (Programmed cell death protein 6, https://www.uniprot.org/uniprot/O75340) interacts with slt (Peptidoglycan lytic exotransglycosylase, https://www.uniprot.org/uniprot/Q8CZP1). HEBP2 is also shown bound to PDCD6 (https://www.rcsb.org/3d-view/5GQQ), which is thought to promote the inhibition of HIV production. b) IGKC (Immunoglobulin kappa constant, https://www.uniprot.org/uniprot/P01834) and yopM (Outer membrane protein, https://www.rcsb.org/structure/1G9U) are interacting, which may inhibit antibody formation. c) IGHA (Immunoglobulin heavy constant alpha 2, https://www.uniprot.org/uniprot/P01877) interacts with tssC (type VI secretion system contractile sheath large subunit, https://www.uniprot.org/uniprot/A0A3N4B420). d) PRDX3 (Thioredoxin-dependent peroxide reductase, https://www.uniprot.org/uniprot/P30048) and mtfA (Involved in the regulation of ptsG expression by binding and inactivating Mlc, https://www.uniprot.org/uniprot/Q7CHU1) with the oligomeric structure of native PRDX3 (https://www.rcsb.org/structure/5UCX). e) CRAT (Carnitine O-acetyltransferase, https://www.uniprot.org/uniprot/P43155) and tssC (S8c Fig) interact with the CRAT native structure (https://www.rcsb.org/structure/1NM8) displayed in structural superposition. f) B2M (Beta-2-microglobulin, https://www.uniprot.org/uniprot/P61769, superposition with the structure of MHC-I: https://www.rcsb.org/structure/1A1M) interacts with YPMT1.34 (Uncharacterized, https://www.uniprot.org/uniprot/O68752). g) YWHAE (14-3-3 protein epsilon, https://www.uniprot.org/uniprot/P62258) and mukB (Chromosome partition protein MukB, https://www.uniprot.org/uniprot/Q8ZG99). h) YWHAZ (14-3-3 protein zeta/delta, https://www.uniprot.org/uniprot/P63104) and istA (IS21-like element IS100 family transposase, https://www.uniprot.org/uniprot/Q7ARN5). i) CSNK2B(Casein kinase II subunit beta, https://www.uniprot.org/uniprot/P67870) and nifj (Putative pyruvate-flavodoxin oxidoreductase, https://www.uniprot.org/uniprot/A0A3N4BEU0) interact. Table A. High-quality predictions from the HPIDB. The number of structures in the

PDB, the number of HP-PPIs in the HPIDB, the number of successful predictions, the number of these where both proteins have an average plDDT > 70 (single chain, sc, plDDT) and high-confidence predictions (sc plDDT > 70 and pDockQ > 0.3) obtained here for human-pathogen protein-protein interactions divided by pathogen.
(PDF)

## Acknowledgments

The computational resources were partly provided by Arne Elofsson (berzelius-2021–29).

## Author contributions

**Conceptualization:** Patrick Bryant.

**Data curation:** Mihkel Saluri, Michael Landreh, Patrick Bryant.

**Funding acquisition:** Michael Landreh, Patrick Bryant.

**Investigation:** Patrick Bryant.

**Validation:** Mihkel Saluri, Michael Landreh.

**Writing – original draft:** Patrick Bryant.

**Writing – review & editing:** Mihkel Saluri, Michael Landreh, Patrick Bryant.

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
