## [Decision Letter · Decision Letter 0]

PCOMPBIOL-D-25-00468

AI-first structural identification of pathogenic protein targets

PLOS Computational Biology

Dear Dr. Bryant,

Thank you for submitting your manuscript to PLOS Computational Biology. After careful consideration, we feel that it has merit but does not fully meet PLOS Computational Biology's publication criteria as it currently stands. Therefore, we invite you to submit a revised version of the manuscript that addresses the points raised during the review process.

Please submit your revised manuscript within 30 days Jun 06 2025 11:59PM. If you will need more time than this to complete your revisions, please reply to this message or contact the journal office at ploscompbiol@plos.org. Please include the following items when submitting your revised manuscript:

We look forward to receiving your revised manuscript.

Kind regards,

Jeffrey Skolnick

Academic Editor

PLOS Computational Biology

Nir Ben-Tal

Section Editor

PLOS Computational Biology

**Additional Editor Comments:**

This is an interesting and worthwhile study which could benefit from minor revision. With regards to the comments of reviewer 1, most are reasonable but it is not necessary to do MD simulations as they will not provide additional insights. Other than this caveat, please address the other concerns of both reviewers in your revised paper.

**Journal Requirements:**

At this stage, the following Authors/Authors require contributions: Mihkel Saluri, Michael Landreh, and Patrick Bryant. Please ensure that the full contributions of each author are acknowledged in the "Add/Edit/Remove Authors" section of our submission form.

4) We do not publish any copyright or trademark symbols that usually accompany proprietary names, eg ©,  ®, or TM  (e.g. next to drug or reagent names). Therefore please remove all instances of trademark/copyright symbols throughout the text, including:

- ® on page: 14

- TM on page: 14.

5) Your manuscript is missing the following section: Discussion.  Please ensure all required sections are present and in the correct order. Make sure section heading levels are clearly indicated in the manuscript text, and limit sub-sections to 3 heading levels. An outline of the required sections can be consulted in our submission guidelines here:

6) Please upload all main figures as separate Figure files in .tif or .eps format. For more information about how to convert and format your figure files please see our guidelines: 

7) We notice that your supplementary Figures, Tables, and information are included in the manuscript file. Please remove them and upload them with the file type 'Supporting Information'. Please ensure that each Supporting Information file has a legend listed in the manuscript after the references list.

8) Please amend your detailed Financial Disclosure statement. This is published with the article. It must therefore be completed in full sentences and contain the exact wording you wish to be published.

9) Your current Financial Disclosure states, "This study was supported by the SciLifeLab & Wallenberg Data Driven Life Science Program (grant: KAW 2020.0239, P.B). Computational resources were enabled by the supercomputing resource Berzelius provided by National Supercomputer Centre at Linköping University and the Knut and Alice Wallenberg foundation with project ids berzelius-2021-29 (Arne Elofsson), Berzelius-2023-267, Berzelius-2024-78 and Berzelius-2024-292 (P.B.). M.L. is supported by a KI faculty-funded Career Position, a Cancerfonden Project grant, a VR Research Environment Grant, a Consolidator Grant from the Swedish Society for Medical Research (SSMF), and the Knut and Alice Wallenberg foundation (2022.0032). "

However, your funding information on the submission form indicates one funder. Please ensure that the funders and grant numbers match between the Financial Disclosure field and the Funding Information tab in your submission form. Note that the funders must be provided in the same order in both places as well.

**Reviewers' comments:**

Reviewer's Responses to Questions

Reviewer #1: This manuscript utilizes AlphaFold-based protein structure prediction methods to predict the structures of host-pathogen protein-protein interactions (HP-PPIs). The authors applied AlphaFold-Multimer (AFM) and FoldDock protocols to predict the structures of 111 curated HP-PPI complexes, comparing TM-scores to determine that FoldDock is more suitable for HP-PPI prediction. Building upon this, the authors predicted the structures of 9,452 HP-PPIs from the HPIDB database using FoldDock, selecting 30 high-quality predictions based on plDDT ≥ 70 and pDockQ ≥ 0.3 for further analysis. In particular, the authors analyzed the complex between human IGKC and Francisella tularensis IPD and observed that IGKC binds at a distinct site from the IPD dimerization interface. Based on this observation, the authors hypothesized that IPD exists as a dimer and interacts with two IGKC molecules, forming a 1:2:1 heterotetramer structure. To validate this hypothesis, the authors conducted native mass spectrometry (nMS) and confirmed the formation of a 1:2:1 complex, demonstrating that AI-based structure prediction can contribute to identifying biologically relevant protein interactions. This manuscript highlights the potential of AI-driven structural modeling in elucidating host-pathogen protein interactions and predicting novel functional interactions. However, there are several issues to be addressed in the current form of the manuscript.

Major comments

1. The authors selected "high-quality predictions" based on structures with an average plDDT ≥ 70 and pTM-score ≥ 0.9. However, it is questionable whether the average plDDT value and pTM-score across the entire protein structure accurately reflect the reliability of the predicted HP-PPI structures. Since this study aims to predict HP-PPI structures, the plDDT of the binding sites and the PAE values between interacting residues are more relevant. However, this aspect seems to be insufficiently addressed.

2. Along the same line with point 1, the authors mention that they found 30 new structures (p. 4, last paragraph). Do they have any experimental evidence to support that those predictions are reliable?

3. The authors mention that pathogen proteins tend to be structurally unstable or contain a high proportion of intrinsically disordered regions (IDRs) (Structure prediction of novel host-pathogen interactions section, second paragraph). It is well known that AlphaFold2 exhibits hallucination effects when predicting secondary structures in IDR regions, which may artificially inflate plDDT values and affect PPI predictions. To address this issue, it would be beneficial to compare the predicted structures using more recent AI tools such as AlphaFold3 or Chai.

4. Did the authors consider HP-PPI interactions in a physiological environment? In vivo, these proteins do not exist in a static state. To better assess the stability of the predicted complexes, molecular dynamics (MD) simulations should be performed, demonstrating that the predicted binding site remains stable throughout the simulation. This would strengthen the credibility of the predictions.

5. The authors may consider comparing the predictions of monomeric and complex forms of the same molecule. This can provide additional insight on conformational flexibility, etc.

6. The interpretation of the predicted complexes in Figure 3 appears questionable. For example, in Figure 3i, the authors suggest that CSNK2B is involved in the Wnt signaling pathway and that its interaction with nifj may disrupt Wnt signaling or contribute to immune evasion. However, this claim is purely based on functional inference and lacks structural evidence. To support this hypothesis, it is necessary to confirm and provide evidence that the functional active site of CSNK2B overlaps with the binding site of nifj.

7. In Figure 4c, the authors report that approximately 25% of IGKC exists as a dimer, based on nMS experimental results. If so, was the IGKC dimer - IPD dimer - IGKC dimer complex observed? Since nMS data alone cannot distinguish between the IGKC dimer - IPD dimer (2:2) complex and the IGKC monomer - IPD dimer - IGKC monomer (1:2:1) complex, if such a structure is possible, the schematic representation should be revised accordingly.

8. More generally, the stoichiometry alone cannot prove that the predicted structure is correct. For example, the authors may suggest a mutation (based on the predicted structure) that can change the stoichiometry and show that it actually works by using MS experiment. If additional data like this are not provided, the current logic is too weak.

Minor comments

1. There are many grammatical errors. I strongly recommend the authors to review the manuscript thoroughly and carefully.

2. The authors provide the hyperlinks of PDB and UniProt databases, but it should be avoided. Instead, I would suggest using PDB and UniProt IDs only. The authors may want to organize the hyperlinks into a separate table.

3. The manuscript presents the IPD-IGKC heterotetramer structure, but does not specify confidence metrics. If this structure was also predicted using FoldDock, the corresponding confidence scores should be reported.

4. The format of reference 2 is inconsistent with the other references.

Reviewer #2: Saluri et al. in their manuscript “AI-first structural identification of pathogenic protein targets,” provide support for a subset of host-microbe interactions by performing an AlphaFold-based FoldDock method for structural prediction and then filtering high-confidence structures using author-defined criteria.

Overall claims:

The title and key claims of the paper are somewhat misleading, as the interactions that the authors investigated were already known prior to this analysis. The authors merely added structural prediction onto the previous evidential data.

The authors should highlight Dyer et al. 2010 in which the interaction between a bacterial dihydrolipoyl dehydrogenase and human IGKC was first identified.

The authors mention interface analysis, although this is missing from the results section, as the results are exclusively at the protein level.

It is unclear how the authors conclude that they have a 4-fold expansion in structures, as they provide 30 new structural predictions, but they claim earlier that there are 52 known complex structures.

Methods:

The authors justify using TM-score instead of DockQ by stating that DockQ does not perform well when native structures differ in length from predicted ones. However, they later used pDockQ in their analysis. Since pDockQ is derived from IF-plDDT and DockQ, wouldn’t it also inherit the same limitations?

The paper does not provide ROC analysis for TM-score thresholds below 0.9. TM-score ≥ 0.5 is already considered the same fold; I would like to see ROC curves for TM-score thresholds at 0.5 or 0.6.

Results:

The statement in Figure 2d is confusing: “Figure 2d shows the number of HP-PPIs with known structure (n=10) and the number of high-quality predictions with an expected TM-score of 0.9 (above 70 plDDT and pDockQ 0.3, n=30). None of the interactions with known structures were accurately predicted.” Could the authors clarify whether “known structures” refer to monomers or complexes?

The HPIDB 3.0 database reports 55,505 unique protein interactions, yet the manuscript states 69,787 interactions in the first sentence of the “Human-pathogen PPIs from the HPIDB” section. Where does this discrepancy come from?

The authors claim to investigate a structural protein-protein interaction network, yet there is no actual network analysis presented in the study.

Minor:

Typo in Figure 4: IDP → IPD

Reviewer #3: This paper reports the results of an effort to predict the 3D structure of host-pathogen protein-protein complexes. The predictions are performed using tools developed based on AlphaFold (AF) and AlphaFold-Multimer (AF-M), respectively trained on single protein chains and protein complexes from the PDB, previously developed by some of the authors. The predictions are performed on two datasets: a set of 111 (24) unique host-pathogen complexes with known 3D structure from the PDB and that of unique binary protein interactions of 9452 human proteins with ten different pathogens with unknown protein structures. The latter interactions, detected by various experimental techniques, were extracted from the HPIDB. Using various structure quality/reliability measures, widely used by the community (with preference given to the TM score) the authors managed to correctly predict (recall) a small fraction of the known complexes and propose highly reliable 3D structures for a small fraction of the tested PPI set from HPIDB. The predicted structure of one of the host-pathogen PPIs from HPIDB was experimentally verified using native MS, and based on this experimental data the stoichiometry of the native complex was defined.

General comments.

The idea of applying AF and AF-M methods to obtain reliable information on the 3D structures of host-pathogen protein interactions makes great sense as it provides valuable information on the 3D structure details of the interacting partners that can be used to design vaccines or drugs likely to counter pathogen infections. The work is solid and represents means of advancing drug and vaccine design developments against pathogen infections.

Minor comments: points demanding clarification.

1 Page 7: The fraction of reliably predicted structures (TM-score of 0.9) is rather low for the full set of 111 known structures and likely slightly lower for the subset of 24 structures that are more recent than those AF/AF-M were trained on (not explicitly shown but surmised from the average values). Why is this the case? What fraction of these two sets were determined by cryo-EM. What is the distribution of the resolution of these structures? Were cryo-EM structures les well predicted? Were lower resolution structures less well predicted?

2-Figure 2c: This Figure shows the distribution of the average plDDT for each chain in the predicted complexes from the HPIDB (n=8441) and the PDB (n=111), and the legend mentions that “93 complexes out of 111 where both chains have over 70 plDDT on average for the PDB set (84%) and 3472 out of 7948 for the HPIDB set (44%)”. These numbers are much higher than the fraction of PPI with TM-Scores above 0.9 for PPIs from these two sources. This seems a bit surprising. It would therefore be helpful to provide a plot illustrating the relationship between the TM-Score and plDDT.

3- Figure 2d plots the number of HP-PPIs with known structure (n=10) and with high-quality predictions (above 70 plDDT and pDockQ 0.3, n=30) for each pathogen. This information is provided here for the first time. What does ‘known structure’ mean in this context: The structure of the full complex or that of one of the partners (human or pathogen)?

4-The question raised above is part a wider imperfection of this otherwise nice manuscript, which is, a lack of a master Table, or 2 such Tables (in the Suppl ), which list the for each PPI dataset (PDB and HPIDB,) all the relevant details : resolution + experimental method (X-ray/cryo-EM; AF/MS/2-Hb/other);origin (host pathogen); # of total predictions; # of predictions with TM-scores≥0.9; pDockQ >0.3; and plDDT >70% for both chain and any other info (diff in sequence length and fraction of disorder when known). And for the PPI’s from HPIDB also provide information on availability experimental structure(s). This will make it much easier to build on this work for future studies.

5- The experimental validation of the predicted 3D structure of the IPD-IGKC, complex by native MS is very encouraging. Are there plans to carry out more extensive validations?

**Have the authors made all data and (if applicable) computational code underlying the findings in their manuscript fully available?**

Reviewer #1: Yes

Reviewer #2: Yes

Reviewer #3: **No: ** See my review report concerning the need of providing more detailed information on the precise content of the data, and the prediction results. I don't remember having seen a link to the data used and that produced in this work. So please double check.

PLOS authors have the option to publish the peer review history of their article (what does this mean? ). If published, this will include your full peer review and any attached files.

**Do you want your identity to be public for this peer review?** For information about this choice, including consent withdrawal, please see our Privacy Policy .

Reviewer #1: No

Reviewer #2: No

Reviewer #3: No

**Figure resubmission:**
---

## [Decision Letter · Decision Letter 1]

Dear Dr Bryant,

We are pleased to inform you that your manuscript 'AI-first structural identification of pathogenic protein target interfaces' has been provisionally accepted for publication in PLOS Computational Biology.

Please notice a few minor reviewer comments. These can be handled while preparing the final draft of the manuscript.

Best regards,

Jeffrey Skolnick

Academic Editor

PLOS Computational Biology

Nir Ben-Tal

Section Editor

PLOS Computational Biology

Reviewer's Responses to Questions

**Comments to the Authors:**

Reviewer #1: The comments are uploaded as an attachment.

Reviewer #3: The authors have addressed all my comments and most of those raised by the 2 other reviewers.

**Have the authors made all data and (if applicable) computational code underlying the findings in their manuscript fully available?**

Reviewer #1: Yes

Reviewer #3: None

PLOS authors have the option to publish the peer review history of their article (what does this mean? ). If published, this will include your full peer review and any attached files.

**Do you want your identity to be public for this peer review?** For information about this choice, including consent withdrawal, please see our Privacy Policy .

Reviewer #1: No

Reviewer #3: **Yes: ** Shoshana J. Wodak

---

## [Editor Report · Acceptance letter]

PCOMPBIOL-D-25-00468R1

AI-first structural identification of pathogenic protein target interfaces

Dear Dr Bryant,

I am pleased to inform you that your manuscript has been formally accepted for publication in PLOS Computational Biology. Your manuscript is now with our production department and you will be notified of the publication date in due course.

With kind regards,

Olena Szabo
